# Cardiovascular Biomarker Profiles in Obesity and Relation to Normalization of Subclinical Cardiac Dysfunction after Bariatric Surgery

**DOI:** 10.3390/cells11030422

**Published:** 2022-01-26

**Authors:** Sanne M. Snelder, Nadine Pouw, Yaar Aga, Manuel Castro Cabezas, L. Ulas Biter, Felix Zijlstra, Isabella Kardys, Bas M. van Dalen

**Affiliations:** 1Department of Cardiology, Franciscus Gasthuis & Vlietland, 3045 PM Rotterdam, The Netherlands; s.snelder@erasmusmc.nl (S.M.S.); y.aga@franciscus.nl (Y.A.); 2Department of Clinical Chemistry, Franciscus Gasthuis & Vlietland, 3045 PM Rotterdam, The Netherlands; n.pouw@franciscus.nl; 3Department of Internal Medicine, Franciscus Gasthuis & Vlietland, 3045 PM Rotterdam, The Netherlands; m.castrocabezas@franciscus.nl; 4Department of Surgery, Franciscus Gasthuis & Vlietland, 3045 PM Rotterdam, The Netherlands; u.biter@franciscus.nl; 5Department of Cardiology, Thoraxcenter, Erasmus University Medical Center Rotterdam, Erasmus MC, 3015 GD Rotterdam, The Netherlands; f.zijlstra.1@erasmusmc.nl (F.Z.); i.kardys@erasmusmc.nl (I.K.)

**Keywords:** obesity/obese, bariatric surgery, cardiac dysfunction, biomarkers

## Abstract

Aims: We aimed to gain insight into the underlying pathophysiology of cardiac dysfunction in obesity patients and the improvement of cardiac function after weight loss. Methods: This is a longitudinal study in which 92 cardiovascular biomarkers were measured by multiplex immunoassays in obesity patients without known cardiovascular disease, before and one year after bariatric surgery. Results: Out of 100 eligible patients, 72 patients completed the follow-up. A total of 72 (78%) biomarkers changed significantly. The biomarkers with the highest relative changes represented processes linked mainly to insulin resistance and inflammation. In the patients with persistent subclinical cardiac dysfunction, the baseline values of 10 biomarkers were different from values in patients with normalization of cardiac function. Most of these biomarkers were linked to inflammation or atherosclerosis. Finally, a model was developed to investigate the relationship between changes in the biomarkers and persistent subclinical cardiac dysfunction. Seven biomarkers were retained in this model, mainly linked to inflammation, atherosclerosis, and hypercoagulability. Conclusion: The majority (78%) of cardiovascular biomarkers changed, pointing mainly to modulation of insulin resistance and inflammation. The baseline levels of 10 biomarkers, as well as pre- to post-bariatric surgery changes in seven biomarkers, were related to persistent subclinical cardiac dysfunction after bariatric surgery.

## 1. Introduction

Obesity has reached epidemic proportions globally and the prevalence will continue to increase [1,2]. The risk of heart failure is known to be increased in obesity patients [3], and subclinical cardiac dysfunction is present even in 60% of obesity patients without known cardiovascular disease [4]. Bariatric surgery has proven to be an effective and safe treatment option resulting in significant, long-term weight loss [5,6]. Weight loss and associated metabolic improvement achieved by bariatric surgery have positive impacts on heart morphology even in obesity patients without heart failure [7], and subclinical cardiac dysfunction normalizes in 50% of the patients one year post-bariatric surgery [8]. However, little is known about the pathophysiology behind this improvement, and it remains unknown why in some patients their cardiac function does not normalize.

Currently, an extensive body of evidence is available on the role of circulating proteins in cardiovascular disease. These proteins reflect several biological processes, such as inflammation, atherosclerosis, insulin resistance, and hypercoagulation that also have been hypothesized to play an important role in cardiac dysfunction in obesity patients [9,10]. Moreover, the use of multiplex immunoassays that determine a broad spectrum of biomarkers is gaining momentum in medical science [11]. In the current study, we investigate changes in cardiovascular biomarker profiles one year after bariatric surgery. Herewith, we aim to gain insight into the underlying pathophysiology of cardiac dysfunction in obesity patients and the improvement of cardiac function after weight loss. 

## 2. Methods

### 2.1. Study Design and Study Group

The protocol of the CARDIOBESE study has been described before [12]. In short, this study is a multicentre, prospective study in which 100 obesity patients who were referred for bariatric surgery were enrolled. Patients were included if they were between 35 and 65 years old with a body mass index (BMI) of ≥35 kg/m^2^. Patients with a suspected or known cardiovascular disease were excluded. Bariatric surgery included either a gastric sleeve, gastric bypass, or a minigastric bypass. Patients were seen before and one year after bariatric surgery. The study protocol was approved by the ethics committee and written informed consent was obtained from all participants [12]. All participants underwent a transthoracic echocardiogram and laboratory tests.

The presence or absence of subclinical cardiac dysfunction in the 100 obesity patients of the CARDIOBESE study has been described in detail before [4]. In short, cardiac dysfunction was defined as either a reduced left ventricular (LV) ejection fraction [13], a decreased global longitudinal strain (GLS), diastolic dysfunction [14], ventricular arrhythmia, or an increased BNP or hs Troponin I. Of the predefined studied parameters, a decreased GLS (<17%) was by far the most abundant, in 57 patients; one had diastolic dysfunction without an available GLS, one had a normal GLS but an increased BNP (49 pmol/L, normal value <30 pmol/L), and one had a positive hs Troponin I. One patient with cardiac dysfunction was diagnosed with acromegaly after inclusion and was excluded from further analysis, leaving 59 obesity patients with versus 40 without subclinical cardiac dysfunction. Of these, 40 patients with and 32 patients without subclinical cardiac dysfunction underwent bariatric surgery and completed the one-year follow-up (Figure 1). These patients were included in the current study.

In order to gain insight into the pathophysiology of cardiac dysfunction in obesity, a broad range of biomarkers was determined before and after bariatric surgery in patients with and without subclinical cardiac dysfunction before surgery.

### 2.2. Laboratory Procedures

Non-fasting blood samples were collected before and one year after bariatric surgery. Blood samples were processed and stored at −80 °C within two hours after collection. Biomarker measurements were subsequently performed in one batch. Serum aliquots were thawed and randomly divided amongst three microwell plates. Internal controls were added to each plate. Plates were frozen at −80 °C and shipped on dry ice to Olink Proteomics AB, Uppsala, Sweden. The cardiovascular panel III of the Olink Multiplex platform for biomarkers was used for analysis. The kits are based on proximity extension assay technology, where 92 oligonucleotide-labelled antibody probe pairs are allowed to bind to their respective targets in the sample. The proximity extension assay technique shows exceptionally high specificity and sensitivity [15,16]. The biomarkers are delivered in normalized protein expression units (NPX), which are relative units; therefore, NPX values for 2 different analyses/proteins are not directly comparable. They are expressed on a log2 scale where 1-unit higher NPX value represents a doubling of the measured protein concentrations. NPX were converted into a linear scale: 2^NPX^ = linear NPX. The abbreviations for all 92 biomarkers are listed in Appendix A. 

### 2.3. Transthoracic Echocardiography

Two-dimensional grayscale harmonic images were obtained in the left lateral decubitus position using a commercially available ultrasound system (EPIQ 7, Philips, Best, the Netherlands), equipped with a broadband (1–5 MHz) X5–1 transducer. All acquisitions and measurements were performed according to current guidelines [13,14]. 

To optimize speckle-tracking echocardiography, apical images were obtained at a frame rate of 60 to 80 frames/s. Three consecutive cardiac cycles were acquired from all apical views. Subsequently, these cycles were transferred to a QLAB workstation (version 10.2, Philips, Best, the Netherlands) for off-line speckle-tracking analysis. The peak regional longitudinal strain was measured in 17 myocardial regions and a weighted mean was used to derive global longitudinal strain (GLS). 

### 2.4. Statistical Analysis

Patients who completed the one-year follow-up were included in the analysis. The distributions of the variables were tested for normality by the Shapiro–Wilk test. Continuous variables with normal distributions were expressed as mean ± standard deviation, those with skewed distributions as median and interquartile range, and categorical variables as counts and percentages. Missing values were omitted (between 0–2%, none of the biomarkers were missing). To compare variables pre- and one year post-surgery, paired *t*-tests were used for continuous variables with normal distributions, the nonparametric Wilcoxon signed-rank test for variables with non-normal distributions, and the McNemar test for categorical variables. 

Relative changes in all biomarkers from pre- to one year post-bariatric surgery were calculated by subtracting the median value of the biomarkers pre-surgery from the value of the biomarkers post-surgery, and dividing the obtained difference by the median value of the biomarkers pre-surgery. In addition to the aforementioned exploration, the change between pre- and post-surgery was analysed by univariable linear mixed modelling for each of the biomarkers, with the moment of measurement (baseline and follow-up) as the independent variable, and all of the biomarkers entered consecutively as the dependent variable. Random intercepts and slopes were used to account for the presence of two biomarker measurements per patient. The Benjamini–Hochberg procedure, with a 5% false discovery rate, was used to correct for multiple testing [17]. 

In the subset of obesity patients with pre-surgery cardiac dysfunction, baseline biomarker levels in those with normalization of cardiac function were compared to levels in those with persistent cardiac dysfunction post-surgery with the Mann–Whitney U test. Again, the Benjamini–Hochberg procedure was used to correct for multiple testing. A multiple biomarker model was then constructed to investigate the relationship between changes in biomarkers and persistent cardiac dysfunction post-surgery. In order to select the subset of biomarkers that carries the best predictive value for cardiac dysfunction and, at the same time, to reduce the risk of overfitting (which is especially important in the setting where the number of events is low relative to the number of predictors), elastic net logistic regression was used. An alpha of 0.3 was used and lambda was selected by cross-validation in the “glmnet” package for the optimization of different model arguments. This method combines two established shrinkage methods: ridge regression and lasso regression [18]. The deltas (value post-surgery minus value pre-surgery) of all individual biomarkers were used simultaneously as inputs for this model. The discriminative ability of the resulting model was investigated by calculating the area under the receiver operating characteristic curve (AUC). The odds ratios of the Z-scores were reported. In addition to the elastic net logistic regression model, we performed a sPLS-DA analysis with the “mixomics” package in R. A two-tailed *p*-value of <0.05 was considered statistically significant, unless otherwise reported. Statistical analyses were performed with SPSS version 25.0 or higher (SPSS Inc., Chicago, IL, USA) or R 3.0.3 (glmnet R package).

## 3. Results

### 3.1. Changes in Clinical Characteristics from before to One Year after Bariatric Surgery 

There was a significant decrease in weight, BMI, systolic blood pressure, and heart rate post-bariatric surgery (Table 1). Moreover, the prevalence of comorbidities such as diabetes mellitus, hypertension, obstructive sleep apnoea syndrome, and the use of medication decreased. 

### 3.2. Changes in Cardiovascular Biomarker Levels from before to One Year after Bariatric Surgery

The relative changes in all 92 biomarkers from before to one year after bariatric surgery are displayed in Figure 2 and in Appendix A. A total of 72 (78%) biomarkers were significantly different: 52 (56%) biomarkers decreased and 20 (22%) increased after bariatric surgery. The biomarkers with the highest relative changes were insulin-like growth factor-binding protein 1 (IGFBP-1) (increase of 175%, *p* < 0.001), integrin beta-2 (ITGBP2) (increase of 139%, *p* < 0.001), epithelial cell adhesion molecule (Ep-CAM) (increase of 90%, *p* < 0.001), osteopontin (OPN) (increase of 59%, *p* < 0.001), N-terminal prohormone brain natriuretic peptide (NT-proBNP) (increase of 58%, *p* = 0.01), and insulin-like growth factor-binding protein 2 (IGFBP-2) (decrease of −58%, *p* < 0.001).

### 3.3. Comparison of Baseline Values of Biomarkers in Patients with versus without Normalization of Cardiac Function after Bariatric Surgery 

Further analysis was performed in the 40 patients with subclinical cardiac dysfunction pre-surgery. Of these 40 patients, one year after bariatric surgery, 20 (50%) had normal cardiac function, and 20 (50%) still had cardiac dysfunction. Table 2 shows that the baseline values of 10 biomarkers were significantly decreased in patients with persistent cardiac dysfunction post-bariatric surgery as compared to patients with normalization of cardiac function: bleomycin hydrolase (BLM hydrolase), caspase-3 (CASP-3), junctional adhesion molecule A (JAM-A), P-selectin (SELP), platelet endothelial cell adhesion molecule (PECAM-1), platelet glycoprotein VI (GP6), platelet-derived growth factor subunit A (PDGF subunit A), retinoic acid receptor responder protein 2 (RARRES2), trem-like transcript 2 protein (TLT-2), and tumour necrosis factor receptor superfamily member 14 (TNFRSF14). 

### 3.4. Association of Changes in Biomarker Levels with Presence of Cardiac Dysfunction Post-Bariatric Surgery 

The elastic net regression model selected the delta of the following set of biomarkers to optimally predict the presence of cardiac dysfunction post-surgery: carboxypeptidase B (CPB1), CASP-3, SELP, GP6, PDGF subunit A, TLT-2, and von Willebrand factor (vWF) (Table 3). Figure 3 shows the ROC curve for this model. The ability of this model to identify patients with cardiac dysfunction post-surgery was high, as shown by the AUC of 0.91 (95% CI: 0.82–0.99, *p* < 0.001). The sensitivity of this model was 90%, specificity 80%, positive predictive value 82%, and negative predictive value 89%. An sPLS-DA analysis was performed in addition to the elastic net regression model. This model largely corresponds to the elastic net model (Appendix A). 

## 4. Discussion

A multiplex immunoassay was used for the first time to investigate changes in a broad spectrum of cardiovascular biomarkers in obesity patients from pre- to one year post-bariatric surgery. The main findings are that the majority (78%) of the cardiovascular biomarkers changed, and reduced levels of 10 biomarkers pre-surgery were related to persistent subclinical cardiac dysfunction post-surgery. Furthermore, a multivariable model showed that changes in seven biomarkers were associated with a lack of improvement in cardiac function.

A total of 72 biomarkers significantly changed from pre- to post-surgery, indicating alterations in a wide range of processes related to metabolic status and cardiovascular function. However, the biomarkers with the highest relative changes mainly represented processes linked to insulin resistance and inflammation. For example, IGFBP-1 is known to be lower in patients with impaired glucose tolerance [19]. IGFBP-1 increased after bariatric surgery, suggesting improved glucose tolerance. Moreover, circulating IGFBP-2 levels are associated with reduced insulin sensitivity in obesity patients [20], and the decrease in IGFBP-2 post-surgery indicates an increase in insulin sensitivity. ITGBP2 is of crucial importance for leukocyte trafficking and immune cell activation, but interestingly plays a role in immune suppression as well. Consequently, dysfunctional or absent ITGPB-2 is linked not only to immune deficiency disease but also to inflammatory disease, thereby contributing to both ends of the spectrum of immune-related diseases [21]. The increase of ITGBP2 post-bariatric surgery may indicate a change in balance towards a decrease of inflammation; however, further research is needed to explore this finding. 

OPN showed a relatively large increase post-bariatric surgery. At first, this was a surprising finding since OPN has been suggested to play a key role in linking obesity to the development of insulin resistance by promoting inflammation [22,23]. Nevertheless, our result is in line with findings from other studies [24,25]. Changes in bone metabolism have been suggested as potential sources of enhanced OPN concentrations post-bariatric surgery, and not inflammation nor insulin resistance [24]. Again, further research will be needed to explore this relationship. 

NT-proBNP also strongly increased post-bariatric surgery. NT-proBNP is known to be decreased in obesity patients, both with and without heart failure [26]. Although the reason for this remains incompletely understood, it is most likely due to lower release in obesity patients, rather than an increase in clearance [27].

A distinctive aspect of the CARDIOBESE study is that different diagnostic techniques were used in parallel to simultaneously investigate a variety of cardiac and metabolic changes after bariatric surgery. This design allowed us to correlate changes in cardiovascular biomarkers with (a lack of) improvement in cardiac function after bariatric surgery.

Baseline values of 10 biomarkers were related to persistent cardiac dysfunction post-surgery. Most of these biomarkers are known to be linked to inflammation and/or atherosclerosis. JAM-A plays an important role in leukocyte transmigration and is upregulated on the early atherosclerotic endothelium [28]. PECAM-1 is upregulated in inflammatory conditions [29], and is particularly evident in atherosclerotic lesions [30]. TNFRSF14 is a mediator of atherosclerosis by inducing inflammation [31]. TLT-2 is known to regulate inflammation through the integration of inflammatory signals [32]. RARRES2 has been associated with inflammation, obesity, and the metabolic syndrome [33]. SELP is expressed at the surface of platelets in activated endothelium and mediates atherosclerotic plaque progression [34]. In addition, GP6 has been described as mediating platelet adhesion on atherosclerotic plaque tissues [35], and PDGF subunit A is expressed by macrophages within atherosclerotic lesions [36]. The remaining two biomarkers do not have a clear relationship with either inflammation or atherosclerosis. CASP-3 is activated in the apoptotic cell and is known to be elevated after myocyte injury [37]. The normal physiological role of BLM hydrolase is unknown [38]; however, it has been suggested that BLM hydrolase may play a part in inflammation by regulating the secretion of chemokines [39]. 

Afterward, a model was developed to investigate the relationship between *changes* in the biomarkers from pre- to one year post-surgery and the presence of subclinical cardiac dysfunction post-surgery. The change in seven biomarkers selected by the multivariable model for the persistence of cardiac dysfunction post-surgery suggests that there is an important role for the combination of inflammatory status (reflected by CPB1 [40], TLT-2 [41], and vWF [42]), markers of atherosclerosis (reflected by SELP [34] PDGF subunit A [36], GP6 [35], CASP-3 [37]), and hypercoagulability (reflected by CPB1 [40], SELP [34], GP6 [35], and vWF [42]).

The relationship between a relative lack of improvement in inflammatory status and persistence of cardiac dysfunction after bariatric surgery is in line with our previously mentioned finding that pre-bariatric surgery values of biomarkers related to inflammation were associated with persistence of cardiac dysfunction. Inflammation is known to be increased in obesity patients, and it has been suggested that heart failure with a preserved ejection fraction in these patients is typically the result of systemic inflammation [43]. The increased size of adipocytes plays a decisive role in inflammation because, to the extent that it increases in the adipose tissue, the production of adipocytokines increases, and this triggers a series of inflammation-related pathophysiological processes [44].

Our study suggests that both increased baseline levels of markers of atherosclerosis and an increase of these levels over time may play a part in the persistence of cardiac dysfunction post-surgery. Obesity is a well-known major risk factor for atherosclerotic vascular disease. The exact mechanism behind this remains to be elucidated, but probably there is an important role for increased inflammation [45]. When atherosclerosis causes myocardial ischemia, it can lead to cardiac dysfunction [46]. 

Hypercoagulability was related to the persistence of cardiac dysfunction in obesity patients after bariatric surgery as well. Hypercoagulability has previously been described in obese patients [47]. Possible explanations for this are the actions of adipocytokines from adipose tissue, increased activity of the coagulation factors, decreased activity of the fibrinolytic system, increased inflammation, increased oxidative stress, endothelial dysfunction, and disturbances of lipids and glucose tolerance in association with the metabolic syndrome [47]. Moreover, the presence of platelet activation and hypercoagulability in heart failure has been well documented [48], suggesting that indeed there may be a relationship between hypercoagulability and cardiac dysfunction in obesity patients, as found in our study. 

## 5. Limitations

Some patients included in the CARDIOBESE study did not undergo bariatric surgery because of various reasons, but mostly because they did not receive the psychologist’s approval or because they withdrew themselves from surgery. Incomplete follow-up was mainly because of withdrawal from follow-up. Although the definition of cardiac dysfunction used in the CARDIOBESE study is unconventional, combining parameters assessed by echocardiography, Holter registration, and blood tests, this was chosen to highlight different potential effects of obesity on cardiac function. The assay that was used to determine the biomarkers is designed as a biomarker discovery tool rather than being an approved clinical test. Therefore, the seven biomarkers related to persistent subclinical cardiac dysfunction after bariatric surgery can currently not be used to predict this in daily clinical practice. Furthermore, while for some of the investigated biomarkers extensive evidence on involvement in biological processes is available, this is lacking for other biomarkers. Finally, the findings of the current study should be considered to be exploratory. Although there were clear associations between echocardiography and laboratory findings, it was not possible to establish a cause–effect relationship.

## 6. Conclusions

The present study provides novel data on 92 cardiovascular biomarkers measured in obesity patients before and one year after bariatric surgery. The vast majority of these biomarkers changed one year after bariatric surgery, indicating alterations in a wide range of processes related to metabolic status and cardiovascular function. However, the biomarkers with the highest relative changes mainly represent processes linked to insulin resistance and inflammation. 

The design of this study allowed for the correlation of changes in cardiovascular biomarkers with a (lack of) improvement in cardiac function after bariatric surgery. Most of the biomarkers with baseline levels associated with persistence of cardiac dysfunction are known to be linked to inflammation, while there also appeared to be a relatively important role for subclinical atherosclerosis. The relationship between changes in certain biomarkers and the persistence of subclinical cardiac dysfunction post-surgery again highlighted the importance of inflammation and atherosclerosis, with a potential role for hypercoagulability as well. Inflammatory status is known to have an important role in the induction of both atherosclerosis and hypercoagulability [49,50]. Thus, although cardiac dysfunction in obesity seems to be a heterogeneous disorder, inflammation plays a central part [43,51].

## Figures and Tables

**Figure 1 cells-11-00422-f001:**
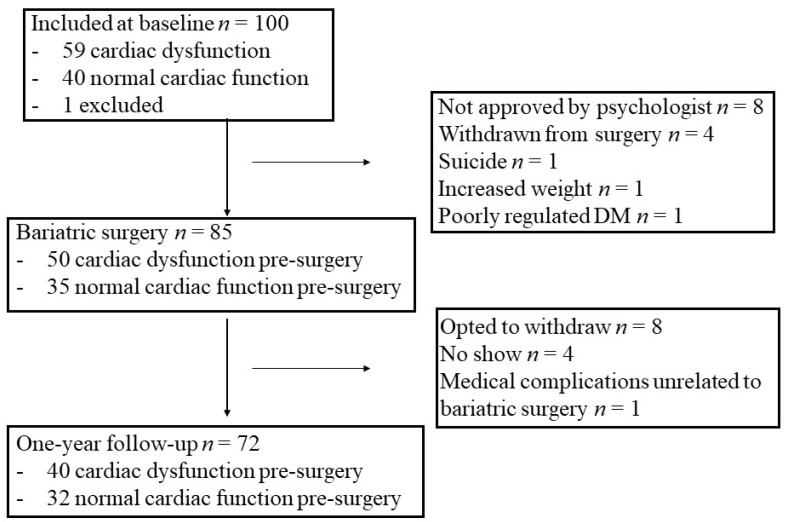
Flow chart of follow-up. DM = diabetes mellitus.

**Figure 2 cells-11-00422-f002:**
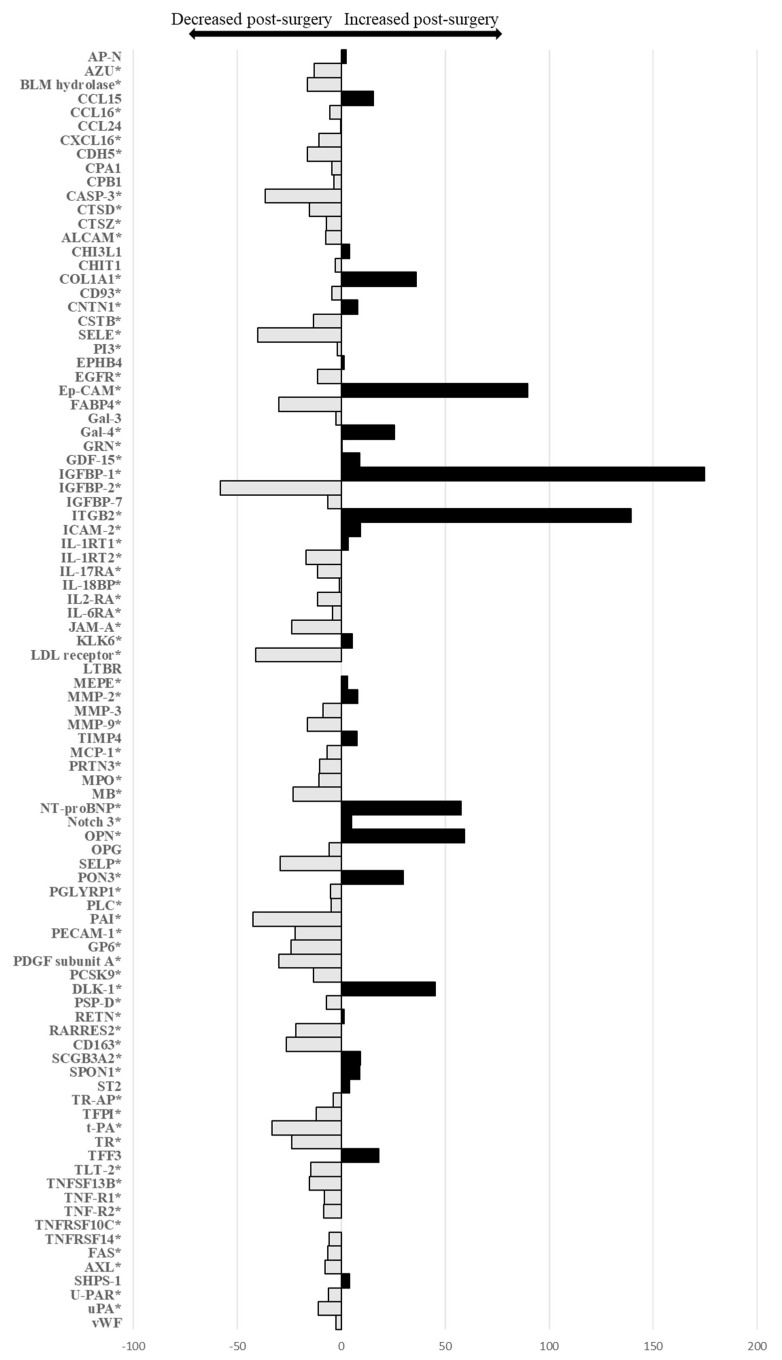
Graphical representation of the relative changes in all 92 biomarkers, pre- and post-bariatric surgery. * significant after Benjamini–Hochberg correction.

**Figure 3 cells-11-00422-f003:**
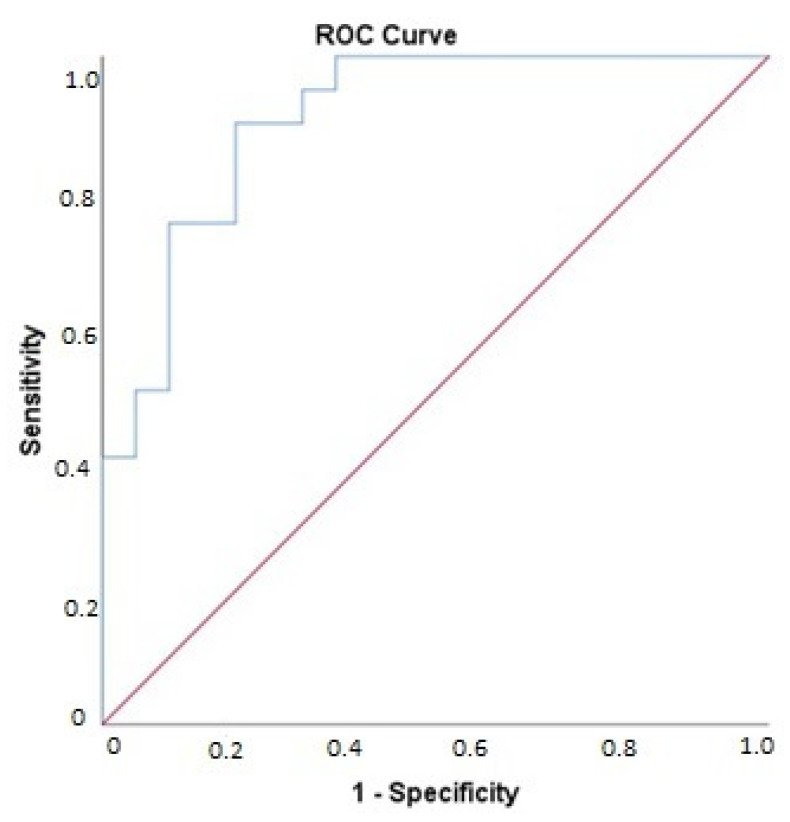
ROC curve for the elastic net model. Biomarkers included are CPB1, CASP3, SELP, GP6, PDGF-subunit A, TLT2, and vWF. AUC 0.91 (95% CI: 0.82–0.99, *p* < 0.001).

**Table 1 cells-11-00422-t001:** Clinical characteristics of the study population. Changes in obesity patients from pre- to post-bariatric surgery.

	Pre-Surgery (*n* = 72)	1 yr Post-Surgery (*n* = 72)	*p*-Value
**General characteristics**			
Age (years)	48 (43–54)		
Female (%)	54 (75%)		
**Physical examination**			
Weight (kg)	122 (113–133)	83 (74–91)	**<0.001**
BMI (kg/m^2^)	41 (39–46)	28 (25–31)	**<0.001**
Systolic BP (mmHg)	146 ± 21	133 ± 20	**0.003**
Diastolic BP (mmHg)	79 (73–88)	80 (75–86)	0.18
Heart rate (bpm)	80 (73–86)	65 (57–71)	**<0.001**
**Comorbidity**			
Diabetes Mellitus (%)	16 (22%)	6 (8%)	**0.002**
Hypertension (%)	24 (33%)	12 (17%)	**0.035**
Hypercholesterolemia (%)	15 (21%)	8 (11%)	0.09
Current smoking (%)	11 (15%)	3 (6%)	0.18
COPD (%)	4 (6%)	0	0.13
OSAS (%)	8 (11%)	0	**0.008**
**Medication**			
Beta blockers (%)	5 (7%)	3 (4%)	0.63
ACE inhibitors/ARBs (%)	11 (15%)	8 (11%)	**0.012**
Calcium channel blockers (%)	6 (8%)	5 (7%)	0.66
Statins (%)	16 (22%)	9 (13%)	**0.039**
Diuretics (%)	13 (18%)	8 (11%)	0.18
Insulin (%)	5 (7%)	4 (6%)	0.56
Oral antidiabetics (%)	10 (14%)	4 (6%)	**0.031**

Values represent mean ± SD, median (Q1–Q3), or *n* (%). *p*-values displayed were analysed by the paired Student’s *t*-test for continuous variables with normal distributions, the Wilcoxon signed-rank test for variables with non-normal distributions, and the McNemar test for categorical variables. Gender, comorbidities, and medication were treated as categorical variables (yes/no). BMI = body mass index, BP = blood pressure, COPD = chronic obstructive pulmonary disease, OSAS = obstructive sleep apnoea syndrome, ACE = angiotensin-converting enzyme, ARBs = angiotensin II receptor blockers.

**Table 2 cells-11-00422-t002:** Comparison of baseline biomarker values in obesity patients with cardiac dysfunction pre-surgery and normalization of cardiac function one year post-bariatric surgery vs. persistent cardiac dysfunction.

Abbreviation	Post-Surgery Normal Cardiac Function (*n* = 20)	Post-Surgery Cardiac Dysfunction (*n* = 20)	*p*-Value
AP-N	37.6 (34.2–41.5)	36.5 (32.6–40.1)	0.81
AZU	7.8 (6.4–10.4)	6.4 (5.3–7.8)	0.61
BLM hydrolase	9.9 (8.2–10.8)	7.4 (6.3–8.6)	0.004 *
CCL15	126 (108–178)	126 (110–183)	0.37
CCL16	155 (114–163)	150 (104–172)	0.97
CCL24	41 (28–68)	43 (23–60)	0.90
CXCL16	51 (40–59)	51 (48–61)	0.15
CDH5	26 (23–28)	26 (19–32)	0.95
CPA1	67 (52–99)	70 (49–98)	0.30
CPB1	61 (46–85)	67 (44–95)	0.31
CASP-3	750 (564–903)	295 (155–551)	<0.001 *
CTSD	8.3 (6.4–11.6)	7.9 (5.9–10.5)	0.67
CTSZ	59.5 (54.5–68.5)	60.1 (43.2–70.6)	0.67
ALCAM	232 (209–284)	222 (197–244)	0.91
CHI3L1	21.1 (17.6–30.0)	18.9 (13.5–27.2)	0.96
CHIT1	26.2 (20.0–38.1)	36.2 (14.9–48.5)	0.84
COL1A1	8.2 (7.3–9.7)	9.1 (8.0–10.8)	0.42
CD93	2200 (2002–2592)	2572 (2058–2955)	0.22
CNTN1	29.1 (24.3–33.5)	27.0 (24.7–32.0)	0.95
CSTB	26.8 (19.9–33.3)	21.5 (17.5–26.5)	0.012
SELE	7543 (5486–9772)	7098 (5587–10,785)	0.38
PI3	5.7 (5.0–7.8)	5.8 (5.0–7.5)	0.41
EPHB4	49.6 (44.8–58.5)	51.4 (46.3–60.4)	0.57
EGFR	11.9 (11.1–13.3)	11.1 (10.1–12.7)	0.22
Ep-CAM	49.6 (33.5–75.1)	51.8 (26.3–126.9)	0.91
FABP4	109.2 (88.9–169.3)	104.2 (85.1–142.8)	0.59
Gal-3	11.5 (10.6–13.1)	11.0 (10.4–12.4)	0.96
Gal-4	19.7 (13.7–23.8)	18.0 (14.8–21.5)	0.72
GRN	60.1 (46.6–75.5)	59.8 (53.6–70.0)	0.42
GDF-15	72.0 (46.6–96.8)	54.4 (48.1–59.4)	0.85
IGFBP-1	10.6 (6.4–18.8)	9.5 (6.2–13.3)	0.63
IGFBP-2	159 (127–200)	170 (133–226)	0.22
IGFBP-7	296 (243–323)	275 (247–322)	0.22
ITGB2	58.4 (49.1–66.6)	54.6 (46.3–65.3)	0.37
ICAM-2	57.3 (48.3–66.9)	53.1 (44.0–69.8)	0.96
IL-1RT1	91.3 (78.1–104.4)	81.9 (74.8–101.4)	0.64
IL-1RT2	57.2 (45.5–62.8)	50.9 (40.9–54.3)	0.91
IL-17RA	24.2 (19.3–33.6)	20.4 (15.8–26.1)	0.06
IL-18BP	72.2 (65.5–80.9)	68.0 (64.6–86.3)	0.65
IL2-RA	15.3 (14.1–17.8)	12.4 (10.2–17.8)	0.033
IL-6RA	5523 (4150–6345)	4812 (3881–6379)	0.96
JAM-A	160 (116–205)	64 (29–103)	<0.001 *
KLK6	5.8 (5.1–7.3)	5.4 (4.6–6.0)	0.26
LDL receptor	27.4 (19.9–40.3)	25.2 (21.6–31.2)	0.82
LTBR	17.9 (15.5–19.4)	17.3 (14.5–19.0)	0.31
MEPE	74.8 (64.0–89.3)	65.7 (62.4–80.6)	0.58
MMP-2	16.6 (14.8–19.6)	18.8 (15.4–20.2)	0.014
MMP-3	183 (137–241)	210 (168–266)	0.018
MMP-9	68.7 (50.8–123.1)	60.5 (36.9–86.8)	0.06
TIMP4	12.3 (11.4–14.7)	14.2 (11.8–15.5)	0.62
MCP-1	20.2 (17.8–24.6)	17.5 (13.9–19.9)	0.33
PRTN3	19.6 (15.7–25.4)	17.4 (14.2–24.9)	0.92
MPO	12.0 (9.8–14.9)	12.0 (9.3–15.0)	0.57
MB	205 (170–267)	226 (193–286)	0.014
NT-proBNP	11.6 (7.2–15.2)	7.7 (4.4–16.6)	0.41
Notch 3	50.2 (41.6–54.4)	53.3 (42.0–63.4)	0.018
OPN	224 (196–271)	218 (165–275)	0.71
OPG	18.0 (14.4–22.6)	16.7 (14.0–19.7)	0.66
SELP	3845 (2941–4907)	2152 (1093–2368)	<0.001 *
PON3	70.7 (61.2–105.7)	91.4 (75.1–106.8)	0.10
PGLYRP1	205 (175–255)	177 (159–211)	0.23
PLC	308 (284–336)	315 (290–348)	0.40
PAI	95 (74–202)	73 (52–97)	0.05
PECAM-1	89 (77–116)	56 (31–63)	<0.001 *
GP6	23 (18–27)	12 (7–16)	<0.001 *
PDGF subunit A	36 (26–48)	19 (13–28)	<0.001 *
PCSK9	9.1 (7.5–11.8)	9.5 (7.7–10.4)	0.57
DLK-1	79 (60–105)	94 (70–105)	0.65
PSP-D	6.4 (5.4–9.9)	8.6 (5.6–10.8)	0.52
RETN	87 (76–108)	81 (71–98)	0.73
RARRES2	4849 (4614–5629)	4256 (4044–4862)	0.003 *
CD163	381 (305–455)	367 (251–407)	0.45
SCGB3A2	5.1 (4.3–6.5)	3.8(3.0–5.2)	0.08
SPON1	4.9 (4.4–5.3)	4.4 (4.0–5.9)	0.73
ST2	33.3 (22.3–40.1)	23.3 (15.3–27.8)	0.047
TR-AP	15.8 (14.4–22.0)	15.8 (13.5–18.3)	0.93
TFPI	792 (688–850)	742 (646–877)	0.50
t-PA	134 (97–167)	144 (96–170)	0.58
TR	65 (39–87)	62 (42–75)	0.70
TFF3	39 (33–46)	38 (28–45)	0.73
TLT-2	73 (64–88)	51 (42–62)	<0.001 *
TNFSF13B	170 (152–184)	174 (143–205)	0.54
TNF-R1	139 (128–155)	141 (128–174)	0.37
TNF-R2	68 (62–78)	65 (61–79)	0.67
TNFRSF10C	192 (132–253)	161 (98–199)	0.45
TNFRSF14	40 (37–46)	33 (29–36)	0.004 *
FAS	79 (73–94)	83 (75–89)	0.17
AXL	650 (547–763)	623 (544–809)	0.29
SHPS-1	16.3 (13.6–20.4)	13.6 (11.5–17.5)	0.62
U-PAR	54.4 (45.6–63.7)	48.8 (39.5–60.7)	0.53
uPA	31.0 (25.9–38.3)	30.7 (25.0–34.3)	0.36
vWF	211 (151–313)	176 (108–239)	0.85

Values represent median (Q1–Q3) of pre-surgery biomarker levels, all units are NPX. *p*-values displayed were obtained with the Mann–Whitney U test. * Significant after Benjamini–Hochberg correction.

**Table 3 cells-11-00422-t003:** Odds ratios of the Z-scores of the biomarkers selected by the elastic net regression.

Biomarker	Odds Ratio
CPB1	0.94
CASP3	1.06
SELP	1.01
GP6	1.12
PDGFsubunitA	1.03
TLT-2	1.22
vWF	1.03

## Data Availability

The data presented in this study are available on request from the corresponding author. The data are not publicly available due to privacy restrictions.

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
