# Peer review of "Cardiovascular Biomarker Profiles in Obesity and Relation to Normalization of Subclinical Cardiac Dysfunction after Bariatric Surgery"

_cells, 2022, doi:10.3390/cells11030422_

Round 1

Reviewer 1 Report

The study presented by the authors is of great importance to investigate sensitive factors associated with persistent cardiovascular outcomes one year after bariatric surgery. The methodologies for data processing, normality analyses, and modeling were articulated with great details and proper description. Several circulating factors were selected to develop a model to discriminate the post-surgery cardiac dysfunction from normal in the subgroup with pre-surgery subclinical phenotypes. Overall, the results provided were well supported and can be potentially valuable for the understanding of the impact of bariatric surgery on cardiac function improvement. Some minor issues could be addressed to enhance the paper:

  1. In table 1, each variable was treated as an individual factor in the calculation in Physical examination, comorbidity, and Medication. However, it is not clear what factors were treated as categorical variables and how many levels were defined. It should also provide if age and sex were used as co-variables. Given the feature of these factors, it would be helpful to provide the outcome from multivariable regression analyses.
  2. Fig 2. The authors indicated that “*, significant after Benjamini-Hochberg correction”—no factors were labeled.
  3. Table 2 is very important. Assuming each factor was also calculated as an independent factor and comparisons were done between the two groups, it would be helpful to provide a clear methodology description.
  4. For feature selection in modeling, the author indicated that “all individual biomarkers” were used as input---how about those with significant differences post-surgery (in Table2) as well as known factors to cardiovascular outcomes (age, sex).

minor issues: the truncated line between 191-192.

Author Response

Reviewer 1

The study presented by the authors is of great importance to investigate sensitive factors associated with persistent cardiovascular outcomes one year after bariatric surgery. The methodologies for data processing, normality analyses, and modeling were articulated with great details and proper description. Several circulating factors were selected to develop a model to discriminate the post-surgery cardiac dysfunction from normal in the subgroup with pre-surgery subclinical phenotypes. Overall, the results provided were well supported and can be potentially valuable for the understanding of the impact of bariatric surgery on cardiac function improvement.

We thank the reviewer for the valuable appraisal of our manuscript and the constructive comments.

Some minor issues could be addressed to enhance the paper:

  1. In table 1, each variable was treated as an individual factor in the calculation in Physical examination, comorbidity, and Medication. However, it is not clear what factors were treated as categorical variables and how many levels were defined. It should also provide if age and sex were used as co-variables. Given the feature of these factors, it would be helpful to provide the outcome from multivariable regression analyses.

Table 1 shows the clinical characteristics of the study population. Gender, comorbidities and medication were treated as categorical variables. Only one level was defined (yes/no). We added this to the caption of table 1.

  1. Fig 2. The authors indicated that “*, significant after Benjamini-Hochberg correction”—no factors were labeled.

Thank you for the alertness, these must have disappeared during editing. We have added them again.

  1. Table 2 is very important. Assuming each factor was also calculated as an independent factor and comparisons were done between the two groups, it would be helpful to provide a clear methodology description.

We already described this in the Methods section as following: “In the subset of obesity patients with pre-surgery cardiac dysfunction, baseline biomarker levels in those with normalization of cardiac function were compared to levels in those with persisting cardiac dysfunction post-surgery with the Mann-Whitney U test” (line 129-131). If the reviewer still considers this to be insufficient, than we of course are more than willing to add further explanation. However, in that case it would be helpful to us if the reviewer is willing to specify the kind of description that we should add.

  1. For feature selection in modeling, the author indicated that “all individual biomarkers” were used as input---how about those with significant differences post-surgery (in Table2) as well as known factors to cardiovascular outcomes (age, sex).

We selected all biomarkers in order to avoid selection bias. For clarification we added the following sentence to the method section: “In order to avoid selection bias, all biomarkers were used in the analysis.”

minor issues: the truncated line between 191-192.

This was corrected.

Reviewer 2 Report

The study is interesting. Its main problem is the lack of power, limiting the validity of the conclusions. 
The authors propose an appropriate methodology to propose a final model. 

For a sensitivity analysis, could you propose a method that handles well multidimensional data like yours? I think, for example of the spls-DA method with the mixomics package or the methods with boosting ( https://doi.org/10.1002/bimj.201700067)! You can also use other methods (e.g. function validate with backwards selection and alpha =0.5 from rms package ( Regression Modeling Strategies With Applications to Linear Models, Logistic and Ordinal Regression, and Survival Analysis, Author: Frank E. Harrell, Jr.).

For the analysis with the elastic net regression model, could you give more information about the optimization of different model arguments?

Could you propose confidence intervals for the R.O.'s, for example, using the R package 'selectiveInference'?

For your final model, could you propose a calibration curve in addition to the roc curve (observed probability versus predicted probability with bias estimation)?

Author Response

Reviewer 2

The study is interesting. Its main problem is the lack of power, limiting the validity of the conclusions. 
The authors propose an appropriate methodology to propose a final model. 

We thank the reviewer for the valuable appraisal of our manuscript and the constructive comments.

For a sensitivity analysis, could you propose a method that handles well multidimensional data like yours? I think, for example of the spls-DA method with the mixomics package or the methods with boosting ( https://doi.org/10.1002/bimj.201700067)! You can also use other methods (e.g. function validate with backwards selection and alpha =0.5 from rms package ( Regression Modeling Strategies With Applications to Linear Models, Logistic and Ordinal Regression, and Survival Analysis, Author: Frank E. Harrell, Jr.).

We performed a spls-DA analysis with the mixomics package as you suggested. This model largely corresponds to our elestic net (5 of the 7 biomarkers are the same). We added the following sentence to the methods section: ”In addition to the elastic net logistic regression model, we performed a spls-DA analysis with the “mixomics” package in R.” And to the results section:” A spls-DA analysis was performed in addition to the elastic net regression model. This model largely corresponds to the elastic net model (supplementary Table 3).”

For the analysis with the elastic net regression model, could you give more information about the optimization of different model arguments?

For optimization of different model arguments we selected an alpha of 0.3 and lambda was selected by cross-validation in the glmnet package in R. We added the following sentence to the methods section: “An alpha of 0.3 was used and lambda was selected by cross-validation in the “glmnet” package for the optimization of different model arguments.”

Could you propose confidence intervals for the R.O.'s, for example, using the R package 'selectiveInference'?

Thank you for the suggestion, unfortunately we were not able to propose the correct confidence intervals for the R.O.’s by using the package selectiveInference. If the reviewer still want us to provide the confidence intervals, we can use a logistic regression model.

For your final model, could you propose a calibration curve in addition to the roc curve (observed probability versus predicted probability with bias estimation)?

We focused only on the selection of the biomarkers with the aim of learning more about the pathophysiology of cardiac dysfunction in obesity patient. With the relatively small dataset, we do not think it appropriate to propose a solid prediction model. Therefore we think that it is not suitable to propose a calibration curve. However, if the reviewer still would like us to a add a calibration curve than we will surely reconsider that.

Reviewer 3 Report

In this manuscript the authors describe a preliminary evaluation of some cardiovascular biomarkers alterations associated with the persistence or not of some cardiac dysfunctions after one-year from bariatric surgery in obese patients.

Although the idea to identify novel prognostic biomarkers to identify patients that have significant possibilities of undergoing cardiac issues after bariatric surgery is very interesting and promising, I believe that this study has many limitations, as already highlighted by the authors themselves. In particular, the number of patients that have been included in the study is very low to be able to define some diagnostic or prognostic biomarkers. Moreover, the data obtained by the immunnoassay weren't validated by any other methodology such as proteomics or more easily with western blot. Therefore, I suggest to improve these preliminary data and try to find some cause-effect relationships that could justify the alterations observed in the patients.

Author Response

Reviewer 3

In this manuscript the authors describe a preliminary evaluation of some cardiovascular biomarkers alterations associated with the persistence or not of some cardiac dysfunctions after one-year from bariatric surgery in obese patients.

Although the idea to identify novel prognostic biomarkers to identify patients that have significant possibilities of undergoing cardiac issues after bariatric surgery is very interesting and promising, I believe that this study has many limitations, as already highlighted by the authors themselves. In particular, the number of patients that have been included in the study is very low to be able to define some diagnostic or prognostic biomarkers. Moreover, the data obtained by the immunnoassay weren't validated by any other methodology such as proteomics or more easily with western blot. Therefore, I suggest to improve these preliminary data and try to find some cause-effect relationships that could justify the alterations observed in the patients.

Thank you for the valuable suggestion. As you mentioned, our study has several limitations that were described in the Limitations section. With our current data, it was not possible to establish a cause-effect relationship between echocardiography and laboratory findings (last sentence Limitations section). Furthermore, indeed, so far we have not been able to validate our data with either proteomics or a western blot. However, as described, our study design was chosen in order to obtain a global impression of the possible pathophysiology of cardiac dysfunction in obesity. We hope and believe that we have managed to do that, but definitely take the suggestion of the reviewer into account and continue our research.  For now, we decided to leave the Limitations section as it is, but if the reviewer still would like us to a add something than we will surely reconsider that.

Reviewer 4 Report

This is a longitudinal study in obesity patients which 92 cardiovascular biomarkers were measured to understand the pathophysiology of cardiac dysfunction in obesity patients after weight loss bariatric surgery. I like to give the following comments.

  1. The cardiovascular biomarkers were not conducted in introduction.
  2. In Figure 2, “Decreased in post-surgery and Increase in post-surgery” will be better to indicate in clear.
  3. Significant changes in 7 biomarkers shall be discussed in detail for each.
  4. The importance of inflammation in addition to hypercoagulability and atherosclerosis mentioned in the conclusion. Please add references to support it.
  5. In conclusion, inflammation plays a central part that needs reason(s) and/or evidence from references.
  6. In limitations, the obtained 7 biomarkers belonged to reliable indicator for diagnosis in clinical practice?

Author Response

Reviewer 4

This is a longitudinal study in obesity patients which 92 cardiovascular biomarkers were measured to understand the pathophysiology of cardiac dysfunction in obesity patients after weight loss bariatric surgery. I like to give the following comments.

We thank the reviewer for the appreciated evaluation of our manuscript and the constructive comments.

The cardiovascular biomarkers were not conducted in introduction.

We added the following sentence to the introduction section: “These proteins reflect several biological processes, such as inflammation, atherosclerosis, insulin resistance and hypercoagulation that also have been hypothesized to play an important role in cardiac dysfunction in obesity patients”.

In Figure 2, “Decreased in post-surgery and Increase in post-surgery” will be better to indicate in clear.

For clarification we have used different colours for decreased (grey) and increased (black).

Significant changes in 7 biomarkers shall be discussed in detail for each.

The role of these 7 biomarkers is further described in the discussion in line 258-263. If this is not sufficient, we are willing to explain a biomarker more detailed.

The importance of inflammation in addition to hypercoagulability and atherosclerosis mentioned in the conclusion. Please add references to support it.

We added the following references to the conclusion:

  1. Rocha, V.Z. and P. Libby, Obesity, inflammation, and atherosclerosis. Nat Rev Cardiol, 2009. 6(6): p. 399-409.
  2. Paulus, W.J. and C. Tschope, A novel paradigm for heart failure with preserved ejection fraction: comorbidities drive myocardial dysfunction and remodeling through coronary microvascular endothelial inflammation. J Am Coll Cardiol, 2013. 62(4): p. 263-71.

In conclusion, inflammation plays a central part that needs reason(s) and/or evidence from references.

We added the following references to the conclusion:

  1. Packer, M., Do most patients with obesity or type 2 diabetes, and atrial fibrillation, also have undiagnosed heart failure? A critical conceptual framework for understanding mechanisms and improving diagnosis and treatment. Eur J Heart Fail, 2020. 22(2): p. 214-227.
  2. Murphy, S.P., R. Kakkar, C.P. McCarthy, and J.L. Januzzi, Jr., Inflammation in Heart Failure: JACC State-of-the-Art Review. J Am Coll Cardiol, 2020. 75(11): p. 1324-1340.

In limitations, the obtained 7 biomarkers belonged to reliable indicator for diagnosis in clinical practice?

Indeed the obtained 7 biomarkers can currently not be used for diagnosis. Therefore we added the following sentence to the limitation section: “ Therefore, the 7 biomarkers related to persistent subclinical cardiac dysfunction after bariatric surgery can currently not be used to predict this in daily clinical practice.”

Round 2

Reviewer 2 Report

The authors have responded to all comments and suggestions. The manuscript is much improved.

Reviewer 3 Report

The authors did not address the major concern about their work but they just highlighted that it should be considered as a proof of concept with many limitations. I hope they will improve their data in the next future to increase the strength of their conclusions.